# Micropropagation and Phytochemical Characterization of *Artemisia ludoviciana* Nutt.: Antioxidant Activity and Phenolic Profiles

**DOI:** 10.3390/plants14243781

**Published:** 2025-12-11

**Authors:** José Miguel Fernández-Cortés, Andrea Amy Balderas-Robles, Elisa Dufoo-Hurtado, Aurea K. Ramírez-Jiménez, Genaro Ruiz-Campos, Alfredo Madariaga-Navarrete, Ashutosh Sharma, Paola Isabel Angulo-Bejarano

**Affiliations:** 1School of Engineering and Sciences, Tecnologico de Monterrey, Ave. Eugenio Garza Sada 2501 Sur, Col: Tecnológico, N.L., Monterrey 64700, Mexico; a01208134@tec.mx (J.M.F.-C.); a01706298@tec.mx (A.A.B.-R.); elisa.dufoo@tec.mx (E.D.-H.); aramirezj@tec.mx (A.K.R.-J.); gruizc@tec.mx (G.R.-C.); 2Centro de Investigación en Ciencia y Tecnología de Alimentos, Instituto de Ciencias Agropecuarias, Universidad Autónoma del Estado de Hidalgo, Av. Universidad Km 1, Rancho Universitario, Hidalgo, CP, Tulancingo 43600, Mexico; alfredo_madariaga@uaeh.edu.mx; 3M^2^-R&D Division, The Mind & Matter Group, Querétaro 76170, Mexico

**Keywords:** estafiate, in vitro, tissue culture, secondary metabolites, bioactive compounds

## Abstract

*Artemisia ludoviciana* Nutt. is an important aromatic plant widely used in traditional Mexican medicine for its therapeutic potential. Its medicinal activity is attributed to a wide range of bioactive compounds, including flavonoids. However, overexploitation, habitat loss, climate change and plant diseases threaten its natural populations and diversity. Although the species reproduces both sexually and asexually, conventional propagation methods are often slow, limited by environmental factors, and susceptible to pests and pathogens. Therefore, this research aimed to establish a micropropagation protocol for *A. ludoviciana* and to evaluate the changes in its phenolic composition and antioxidant activity while adapting to ex vitro conditions. Full-strength Murashige and Skoog (MS) media supplemented with 0.1 mg L^−1^ 6-benzylaminopurine resulted in the highest number of shoots (3.30 ± 0.34) and shoot length (3.00 ± 0.12 cm). Moreover, 1/2 MS media supplemented with 0.5 mg L^−1^ indole-3-acetic acid improved the number (14.45 ± 0.56) and quality of roots. Hardening and acclimatization of plantlets showed 100% survival after 10 weeks. Also, the phenolic composition and antioxidant activity of *A. ludoviciana* changed in response to stress derived from growth conditions. The results support the sustainable use and rapid propagation of the species, as well as provide the basis for the study of secondary metabolism in the plant.

## 1. Introduction

*Artemisia ludoviciana* Nutt., also known as “estafiate” or “silver wormwood”, is an aromatic plant species from the Asteraceae family, one of the largest plant families and one of the most important in traditional medicine [1,2]. This herb, native to North America, and distributed from Canada to Guatemala, possesses important ethnomedicinal value [3,4]. The plant has been used in Mexico since pre-Hispanic times, where it was considered to have a divine and healing nature [5]. At present, the decoction of its aerial parts has several applications in Mexican folk medicine, including the treatment of parasites, intestinal colic, diarrhea, reflux, indigestion, vomiting, stomachal infections, gall bladder malfunction, cold, flu, headaches, and diabetes [1,6,7,8,9,10,11,12]. Due to its ethnomedicinal importance, pharmacological and phytochemical research on this plant has resulted in the identification of several bioactive compounds, with monoterpenes, sesquiterpenes, and flavonoids being the most relevant [13]. The essential oils of *A. ludoviciana* possess pharmacological activities, such as antimicrobial [14], antinociceptive [15], and antidiarrheal [16] activities. In addition, *A. ludoviciana* extracts have shown antimicrobial, antiparasitic, anti-inflammatory [17,18] antinociceptive [19], hypoglycemic [20] and antioxidant [21] properties. However, these phytochemical and pharmacological characteristics may change among populations, due to genotypic diversity, ecotypic variation, and developmental stage [22,23].

The lack of control over the collection and geolocation of plant material may affect the plant’s bioactive compounds, and consequently, its medicinal properties [24]. In addition, poor use and harvesting practices could lead to its extinction. Although *A. ludoviciana* is not listed as endangered, overexploitation, habitat destruction, climate change, and plant diseases threaten the genetic diversity and existence of this and many other medicinal plants [25,26]. Conversely, they can be protected through in situ (within their natural habitats) or ex situ (outside their natural habitats) conservation techniques, such as plant micropropagation. Combined with strong legal and policy frameworks, these strategies ensure the availability of genetic resources for research, cultivation, and sustainable use [26]. Although *A. ludoviciana* reproduces both asexually and sexually, it is commonly propagated vegetatively by rhizomes [27]. However, this method, like other conventional techniques, has its limitations, such as low multiplication rates, environmental constraints, and susceptibility to pathogens. Compared to these approaches, micropropagation allows rapid, large-scale, and disease-free propagation of plants [28]. Furthermore, this tissue culture technique supports the natural conservation, growth, and use of plant resources and is also considered an important tool for plant research [29]. The present study aims to establish a rapid and reproducible micropropagation protocol for *A. ludoviciana* through in vitro axillary shoot proliferation and rooting, followed by hardening and acclimatization under ex vitro conditions. In addition, the phenolic composition and antioxidant activity were analyzed at different time intervals during ex vitro development and compared with those of the mother plant, to evaluate the phytochemical profile of propagated plants and assess changes in bioactive compound content during the development of *A. ludoviciana*.

## 2. Results

### 2.1. Culture Initiation

Low contamination was achieved using the disinfection treatment described previously. This treatment reduced contamination by increasing the exposure time of explants to the 10% (*v*/*v*) sodium hypochlorite solution and a 0.1% (*v*/*v*) Tween-20^®^ solution from 15 to 20 min. After ensuring the axenic culture in Murashige and Skoog (MS) basal media, 6-benzylaminopurine (BAP) was tested in low concentrations to allow the shoots to develop and acquire enough material for shoot proliferation. Shoot induction in MS media containing 0.05 mg L^−1^ BAP improved the number and quality of induced shoots (Figure 1B,C).

### 2.2. Shoot Proliferation

After 4 weeks of culture in MS media containing different BAP and KIN (Kinetin) concentrations, all treatments (T0–T6) led to shoot formation (Table 1).

Shoot formation was observed on the fourth day after culture, and BAP (T1–T3) and KIN (T4–T6) significantly increased the number of shoots compared to the control (T0). The highest number of shoots per explant (3.30 ± 0.34) and the greatest average shoot length (3.00 ± 0.12 cm) were obtained at 0.1 mg L^−1^ BAP (T1) (Figure 1E,F). As the concentration of BAP increased (0.5 and 1.0 mg L^−1^ BAP), the number and length of shoots decreased compared to T1. In contrast, an increment in KIN concentrations resulted in a higher number of shoots between T4 and T5, although there was no significant increase between T5 and T6. Additionally, T5 significantly improved shoot length compared to the other KIN treatments.

### 2.3. Root Induction

Root formation was observed in all treatments (R0–R6) after 6 days of culture in root induction media. After 2 weeks, the media supplemented with α-naphthaleneacetic acid (NAA) (R1–R3) and indoleacetic acid (IAA) (R4–R6) significantly increased the number of roots per shoot, compared to the control (R0) (Table 2).

Lower IAA concentrations resulted in higher root formation compared to NAA, where root number increased with increasing concentrations. The highest root induction was obtained at 0.5 mg L^−1^ IAA (R5), with 14.45 ± 0.56 roots per shoot (Figure 1G). Although R5 produced significantly shorter roots (2.85 ± 0.05 cm) compared to those in other treatments, it visually promoted the development of a well-formed root system with higher secondary root induction. Conversely, NAA concentrations resulted in greater root elongation than IAA, with 0.5 mg L^−1^ NAA (R2) being the most effective for root elongation (4.04 ± 0.06 cm) compared to the other rooting treatments. Additionally, higher concentrations of auxins negatively impacted the shoot growth, while lower concentrations allowed better shoot development (R1 and R4).

### 2.4. Hardening, Acclimatization, and Reproducibility

After the in vitro culture, the plantlets were hardened and successfully acclimatized (Figure 1H and Figure 2). During acclimatization, the plants were transferred to larger pots and maintained under minimum controlled conditions, with irrigation as the only controlled factor. As the plant grew under ex vitro conditions, notable morphological changes were observed, including changes in leaf color, development of foliar tomentum, and progressive hardening of the stem (Figure 2). After 2 months of acclimatization, all plantlets demonstrated a 100% survival rate, showed no morphological variation, and had uniform growth.

To test reproducibility, a second *A. ludoviciana* population was micropropagated following the established methodology (Figure 3). Morphological differences were observed between both populations before, during, and after running the protocol (Figure 3A–F). However, the second population responded positively to the treatments and was successfully hardened and acclimatized, with no visible abnormalities in their morphology.

### 2.5. Total Phenolic and Flavonoid Content and Antioxidant Capacity

As shown in Table 3, the total phenolic composition (TPC) varied throughout the ex vitro acclimatization process. The lowest TPC was recorded at D7 (1.34 ± 0.01 g GAE 100 g^−1^ DW), while the highest was obtained at 2M (2.63 ± 0.01 g GAE 100 g^−1^ DW). A marked increase was observed at D14 (2.50 ± 0.01 g GAE 100 g^−1^ DW), followed by a slight decline at 1M (1.80 ± 0.00 g GAE 100 g^−1^ DW); in contrast, the control plant exhibited significantly higher TPC levels compared to the in vitro-derived plants. Total flavonoid content (TFC) showed a similar trend. TFC values were lowest at D7 (0.13 ± 0.06 g QE 100 g^−1^ DW) and highest at 2M (1.38 ± 0.12 g QE 100 g^−1^ DW). The TFC showed a progressive increase between D14 and 2M, suggesting a gradual re-establishment of secondary metabolism under acclimatized conditions.

Antioxidant capacity, as measured by radical scavenging activity (RSA), peaked at 2M with an inhibition of 81.92 ± 0.50% and remained high compared to early hardening stages. The highest RSA was recorded at 1M (82.50 ± 0.79%), while the lowest was observed at D7 (38.21 ± 1.08%). Similarly, Trolox equivalent antioxidant capacity (TEAC) values increased steadily from D7 (58.51 ± 2.29 µmol TE 100 g^−1^ DW) to 2M (118.51 ± 1.35 µmol TE 100 g^−1^ DW), the highest antioxidant capacity recorded. These results indicate that both phenolic biosynthesis and flavonoid biosynthesis, as well as antioxidant potential, are progressively restored during ex vitro development, with the most substantial biochemical recovery observed by 2 months of acclimatization.

The correlation between total phenolic content, total flavonoids, and antioxidant activities of the extracts of *A. ludoviciana* samples was evaluated using Pearson’s correlation test (Table 4). There was a positive relationship between all data, but it was not significant in all cases. A significant correlation was found between the total flavonoid content and the total phenol content (r^2^ = 0.942). The TFC also had a significant correlation with the total equivalent antioxidant capacity and radical scavenging activity of extracts, with correlation coefficients of r^2^ = 0.974 and r^2^ = 0.841, respectively.

### 2.6. Determination of Phenolic Composition

A high-performance liquid chromatography (HPLC) coupled with diode-array detector (DAD) analysis allowed the detection of ten major peaks among *A. ludoviciana* samples, each peak corresponding to a single compound (Table 5). From the total number of peaks observed, only two were identified by the available standards. Spiking of the standards allowed the identification of peaks 3 and 7 as caffeic acid and ferulic acid, respectively.

Caffeic acid was detected from 2-week ex vitro plant extracts (D14) at a concentration of 0.20 ± 0.02 mg g DW^−1^ and increased to 0.56 ± 0.01 mg g DW^−1^ after two months (2M) of acclimatization. This value was significantly higher than the concentration of caffeic acid in the control plant (MP; 0.28 ± 0.02 mg g DW^−1^). Furthermore, ferulic acid was detected in all tested samples at different stages, but its concentration was significantly lower in extracts from in vitro plants (IP; 1.94 ± 0.19 mg g DW^−1^) and those collected after 7 days of hardening (D7; 2.15 ± 0.12 mg g DW^−1^). This concentration increased significantly to 4.46 ± 0.30 mg g DW^−1^ in 2-week ex vitro plant samples and remained constant throughout the acclimatization process, reaching a value of 4.63 ± 0.11 mg g DW^−1^. Although the control plant showed a higher concentration of 5.75 ± 0.26 mg g DW^−1^, this difference was not statistically significant (*p* > 0.05).

Regarding the unidentified compounds, peaks 2, 8, and 9 were among the most abundant in the samples. The concentration of peak 2 followed a similar pattern to that of the total phenolic content. It reached 3.93 ± 0.59 mg g DW^−1^ on D14 but decreased significantly to 2.48 ± 0.08 mg g DW^−1^ in 1-month-acclimatized samples (1M). Peak 8, like caffeic acid, was detectable at D14 with a concentration of 0.13 ± 0.02 mg g DW^−1^ and increased significantly at 2-month-acclimatized plants (2M), reaching 1.43 ± 0.11 mg g DW^−1^. However, this concentration was significantly lower than that reported in the control plant (2.27 ± 0.25 mg g DW^−1^). In contrast, the concentration of peak 9 was significantly variable across samples, with no trend observed. However, it was significantly higher at 1M, with a concentration of 10.34 ± 0.09 mg g DW^−1^, compared to the other plant samples and the control plant.

## 3. Discussion

The genus *Artemisia* L. is one of the most diverse, extensive, and widely distributed of the Asteraceae family [29,30]. It comprises more than 500 species, found mainly in temperate regions of Asia, Europe, and North America [29]. Many of these species are valuable in traditional medicine, as well as in the food industry, ornamentation, and land restoration [31]. However, overexploitation of the genus, combined with the impacts of climate change, is threatening its biodiversity [32]. Therefore, the development of efficient propagation protocols is important for the sustainable use, study, and conservation of *Artemisia* species, particularly those with medicinal value. Thus, this study focused on the micropropagation and phytochemical analysis of *A. ludoviciana* Nutt., a species with important therapeutic potential [33].

Axillary shoot proliferation was used in this study because, compared to indirect methods, this approach minimizes the risk of genetic variation and thus preserves genetic fidelity [34]. Successful shoot induction and proliferation were achieved using both BAP and KIN. The use alone or in combination of these cytokinins for shoot induction and proliferation has also been reported in other *Artemisia* species [35,36,37]. In this study, BAP showed better results than KIN, which could be related to prolonged stimulation of cell division, as BAP is not easily broken down by plants and is also more stable and resistant to oxidation compared to other cytokinins [38]. Furthermore, the most effective concentration for shoot induction and proliferation was 0.1 mg L^−1^ BAP, while higher levels resulted in fewer shoots per explant and smaller shoot size. This pattern suggests a hormetic effect, indicating a favorable response of the explants when exposed to low cytokinin levels and growth inhibition at higher plant growth regulators (PGR) concentrations [39]. A similar effect was observed in *Artemisia vulgaris* and *Artemisia arborescens* where higher BAP concentrations had an unfavorable effect [35,37]. It has also been reported that higher cytokinin concentrations are related to in vitro disruption and hyperhydricity [40].

In other species of the genus, the use of MS or half-strength MS medium supplemented with auxins promoted direct root formation in shoots [35,36,41]. In this work, all tested treatments induced rooting in the in vitro raised shoots of *A. ludoviciana*. However, IAA was found to be more effective than NAA at lower concentrations, probably due to the chemical nature of each PGR and the way they are recognized and metabolized by the plant’s tissue [42]. Similar results were obtained in the induction of *A. nilagirica* roots, where a half-strength MS medium supplemented with different concentrations of these auxins was also tested [43]. Likewise, IAA gave better results than NAA in the proliferation of roots in *A. vulgaris* [35]. Although auxins may promote root development while inhibiting shoot growth, we observed that small auxin concentrations led to both shoot elongation and higher root number and length in *A. ludoviciana*. Similarly, cytokinin hormesis could be observed in the explants exposed to different auxins. In addition, the use of low PGR concentrations is not only important for ensuring the economic feasibility of micropropagation protocols, but also the genetic stability of micropropagated plants [44]. Somaclonal variation refers to the genetic and phenotypic variation that occurs in plants regenerated through tissue culture and remains a major problem of many in vitro-cultured plants. Several factors could condition the frequency of this variation, including the micropropagation method, genotype, source of the explant, type and concentration of growth regulators, number of subcultures, and duration of culture [45]. In the present study, these factors were kept to a minimum to reduce potential variation, and no evident morphological differences were observed between the mother plants when compared to the in vitro acclimatized plants at the end of the experiment.

Progressive morphological and phytochemical changes were observed in ex vitro plantlets throughout hardening and acclimatization. During this period, in vitro-propagated plantlets are exposed to both biotic and abiotic stresses such as higher temperatures and light intensity, low air relative humidity, and the presence of microorganisms and pathogens [46,47]. Plants undergo different anatomical and physiological changes in response to these conditions, such as an increase in vein density and trichome number, changes in stomata shape and size, and development of the stems’ chlorenchyma and sclerenchyma fibers [48]. Some of these morphological changes have been previously reported in diploid leaves of *A. tridentata* species [49]. On the other hand, the plant’s antioxidant system can be triggered in response to an increase in reactive oxygen species (ROS) content, derived from stress on the plant. This defensive system comprises antioxidant enzymes and low molecular antioxidants, including ascorbate, carotenoids, glutathione, tocopherols, and phenolic compounds [50,51]. Thus, the biochemical response of *A. ludoviciana* to ex vitro conditions was evaluated. The total phenolic and flavonoid content, as well as its antioxidant capacity, changed through this period in response to stress derived from the plant’s growth conditions. The values reported in this study for total phenolic and total flavonoid contents are significantly higher than those previously described for methanolic extracts of *A. ludoviciana* leaves (0.36 ± 0.00 g GAE 100 g^−1^ DW; 0.10 ± 0.01 mg QE 100 g^−1^ DW) [21]. However, these results are comparable to those obtained in extracts from the aerial parts of *A. annua*, *A. absinthium*, *A. austriaca*, *A. pontica*, and *A. vulgaris* [52,53]. Total phenolic content decreased significantly after 1 month of acclimatization and accumulated again in 2-month-acclimatized plants. This is a result of the growth conditions and the developmental stage of *A. ludoviciana*. Conversely, the total flavonoid content accumulated gradually in the plant from day 7 under ex vitro conditions. Similarly, the antioxidant capacity increased progressively as the plant developed, despite the observed changes in the total phenolic content of the plant. Oh et al. [54] reported no correlation between the total phenolic content and the antioxidant capacity of extracts from *A. gmelinii* aerial parts. This could be due to the antioxidant capacity of some phenols, including flavonoids, or to the presence of other non-phenolic compounds such as ascorbic acid, carotenoids, and some terpenoids [55,56]. In this research, a strong, significant correlation was found between the equivalent total antioxidant capacity and the total flavonoid content of *A. ludoviciana*, suggesting that this class of compounds is directly related to the antioxidant activity of the species.

The HPLC-DAD analysis resulted in the quantification of ferulic and caffeic acid in the samples. Both compounds increase the tolerance of plants to stress by scavenging reactive oxygen species, promoting plant growth, increasing the rate of photosynthesis, and regulating the expression of genes encoding stress-related proteins [57,58]. Therefore, these compounds were expected to increase in the plant as it developed under ex vitro conditions. Carvalho et al. [21] reported the phenolic composition of *A. ludoviciana* and five other species of the genus. In this study, caffeic acid and ferulic acid conjugates were the most abundant phenolic compounds of the species. In another research, ferulic acid was identified as one of the most concentrated phenolic acids of the plant, below chlorogenic acid and rosmarinic acid [59]. Moreover, chlorogenic acid and 3,5-dicaffeoylquinic acid have been reported as major constituents in extracts from *A. ludoviciana* [60]. The latter, along with other flavonoids such as eupatilin and jaceosidin, contribute to the hypoglycemic activity of the plant. Some of the unidentified peaks in the chromatograms may correspond to some of these major compounds identified in the species. The differences between previous findings on *A. ludoviciana* and the results reported in this research may be attributed to various factors, including environmental growing conditions, genetic differences, developmental stage, the plant part, or the extraction method used [61,62,63].

## 4. Materials and Methods

### 4.1. Plant Materials

Fully grown plants of approximately 15 cm in length were taken from the Botanical Garden of the Autonomous University of Queretaro (JBUAQ) (100°26′38″ W; 20°42′01″N; 1922 m altitude) in March 2023 and authenticated in the Herbarium Jerzy Rzedowski (QMEX) of the Autonomous University of Queretaro (UAQ) with voucher number QMEX00006862. Mother plants were placed in a tier greenhouse and watered every 3 days.

### 4.2. Medium and Culture Conditions

For all experiments, MS [64] basal medium (PhytoTech Labs, Overland Park, KS, USA) supplemented with 3% (*w*/*v*) sucrose (CTR, Monterrey, NL, Mexico), and 0.8% (*w*/*v*) agar (BD, Franklin Lakes, NJ, USA) was used unless stated otherwise. The media pH was adjusted to 5.7 ± 0.02 with either 1N hydrochloric acid (HCl) or 1N potassium hydroxide (KOH) (Karal, Leon, GTO, Mexico) before autoclaving at 121 °C and 15 lbs pressure for 15 min. After this, ~25 mL of freshly sterilized media was poured into sterile glass culture vessels of 180 mL (50 mm D × 90 mm H) which were sealed with polypropylene caps. Culture conditions were maintained at 25 ± 2 °C under a 12 h light and 12 h dark photoperiod, with a photosynthetically active radiation (PAR) of 100 μmol m^−2^ s^−1^ supplied with white, fluorescent lamps.

### 4.3. Surface Sterilization and Culture Initiation

Shoot segments of approximately 1.5 cm in length, each containing about two axillary buds, were excised, and the leaves were carefully removed with a scalpel to avoid damaging the plant material (Figure 1A). The explants were immediately placed inside the laminar flow hood and surface sterilized by immersion using a 0.5% (*v*/*v*) ionized silver (Microdyn^®^, Santa Catarina, NL, Mexico) solution for 10 min, followed by 10% (*v*/*v*) sodium hypochlorite solution (Clorox^®^) and a 0.1% (*v*/*v*) Tween-20^®^ solution for 15 min. Then, the nodal segments were rinsed thrice with sterile distilled water to remove any solution residue and were cut into 1 cm pieces to be finally cultured on a MS basal medium devoid of plant growth regulators PGRs. After 4 weeks of axenic culture, the explants were subcultured in MS media containing 0.05 mg L^−1^ 6-benzylaminopurine (BAP) (PhytoTech Labs, Overland Park, KS, USA) for shoot induction.

### 4.4. Shoot Proliferation

Axenic axillary shoots (4-week-old) derived from shoot induction media (full-strength MS supplemented with 0.05 mg L^−1^ BAP) were used to study the effect of PGRs on shoot proliferation. The shoots were cut into nodal segments, 1 cm in length, and cultured in MS medium supplemented with BAP or KIN (PhytoTech Labs, Overland Park, KS, USA) at different concentrations (Table 6). The number and height of shoots per explant were recorded after 4 weeks of culture to determine the best shoot proliferation treatment.

### 4.5. Rooting

Four-week-old shoots (1.5 cm in length) derived from the best shoot induction treatment (full-strength MS supplemented with 0.1 mg L^−1^ BAP) were excised and transferred to ½MS medium supplemented with NAA or IAA (PhytoTech Labs, Overland Park, KS, USA) (Table 7). Various PGR concentrations were tested to identify the optimal rooting treatment. Following 2 weeks of incubation, the number and length of roots per explant and shoot length were registered.

### 4.6. Hardening, Acclimatization, and Reproducibility

After 2 weeks of culture, rooted shoots cultured in ½MS media supplemented with 0.5 mg L^−1^ IAA (R5) were carefully removed from the culture vessels and rinsed gently with tap water to remove the residual medium. Then, the plantlets were transferred to 12-cell plastic culture trays (38 mm L × 50 mm H) containing a sterilized soil mixture of peat moss and perlite (4:1 *v*/*v*). The trays were covered with a transparent lid, with the air vent half-open, and placed in a growth room at 25 ± 2 °C under a 12 h light and 12 h dark photoperiod, with photosynthetically active radiation (PAR) of 100 μmol m^−2^ s^−1^ provided with white, fluorescent lamps, and a relative humidity of 52 ± 5%. On the 7th day of incubation, the lid was removed, and after 14 days of ex vitro culture, the plants were transferred to plastic pots (150 mm D × 110 mm H) containing the same sterilized soil mixture. The plants were placed in an open greenhouse and were watered every 2 to 3 days with tap water for acclimatization.

To test the reproducibility of the established micropropagation protocol within the species, a second *A. ludoviciana* population was collected in March 2024 from La Cañada, Queretaro (100°20′11″ W; 20°36′08″ N; 1940 m altitude), and authenticated under the voucher number QMEX00000327. Plants from this population were in vitro-propagated, hardened, and acclimatized using the previously described protocol.

### 4.7. Total Phenolic and Flavonoid Content and Antioxidant Capacity

The phenolic composition of *A. ludoviciana* was assessed during the hardening and acclimatization process. Aerial parts were collected on the last day of in vitro micropropagation (IP), after 7 days (D7) and 14 days (D14) of hardening, and after 4 weeks (1M) and 8 weeks (2M) of acclimatization. The mother plant was considered as control (MP). Plant material was sampled from 12 individuals per experimental period and dried at 40 °C for 48 h, then vacuum-sealed and stored in the dark until further use. The dried samples were milled and extracted with HPLC-grade methanol (Karal, Leon, GTO, Mexico) at a ratio of 1× *g* of dried plant material per 10 mL of solvent (1:10 *w*/*v*) for 20 min at 40 °C in an ultrasonic bath (Cole-Parmer, Vernon Hills, IL, USA). Then the samples were centrifugated (6000 rpm for 10 min). The supernatant was recovered, filtered through 0.45 µm hydrophobic PTFE syringe filters, and stored at −20 °C. To calculate the dry weight percentage, plant material was sampled at each experimental period and weighed after harvest to determine fresh mass, and then oven-dried at 60 °C until constant weight to obtain dry mass.

To determine the TPC of the methanolic extracts, each sample was diluted at a 1:400 with distilled water, in 2 mL Eppendorf tubes, 50 µL of the diluted extract was loaded, followed by the addition of 125 µL of 1N Folin–Ciocalteu (Sigma-Aldrich, St. Louis, MO, USA) reagent and 250 µL of distilled water. The mixture was vortexed for 5 min. Subsequently, 625 µL of a 7% (*w*/*v*) sodium carbonate solution was added. The tubes were incubated at room temperature for 2 h in the dark. After incubation, 200 µL of the reaction was added to a 96-well microplate, and absorbance was measured at 765 nm using an xMark™ Microplate Absorbance Spectrophotometer (Bio-Rad Laboratories, Hercules, CA, USA). Gallic acid (Sigma-Aldrich, St. Louis, MO, USA) was used as the standard, and a calibration curve was prepared using concentrations from 0 to 0.009 mg/mL. Results were expressed as g gallic acid equivalents (GAE) per 100 g of dry weight (DW). This protocol was adapted with some modifications from Wong-Paz et al. [65]

For TFC, a colorimetric method based on the formation of a flavonoid-borate complex was used following the protocol described by Oomah et al. [66]. For the assay, 50 µL of the diluted extract was mixed with 100 µL of methanol and 50 µL of a 1% 2-aminoethoxydiphenil borate (2-APB) (Sigma-Aldrich, St. Louis, MO, USA) solution. The mixture was incubated at room temperature in the dark for 10 min. After incubation, absorbance was measured at 404 nm. Quercetin (Sigma-Aldrich, St. Louis, MO, USA) was used as the standard, and a calibration curve was prepared using concentrations from 0 to 0.08 mg/mL. Results were expressed as g quercetin equivalents (QE) per 100 g of DW.

The RSA and the TEAC of the extracts were evaluated using the 2,2-diphenyl-1-picrylhydrazyl (DPPH) and 2,2′-azino-bis (3-ethylbenzothiazoline-6-sulfonic acid) (ABTS) assays, respectively. For the DPPH assay, in a 96-well microplate, 100 µL of the diluted extract was mixed with 100 µL of a 0.1 mM DPPH (Sigma-Aldrich, St. Louis, MO, USA) methanolic solution. The reaction was incubated for 30 min at room temperature in the dark. After incubation, absorbance was measured at 520 nm. Gallic acid was used as the standard, and a calibration curve was prepared using concentrations from 0 to 70 µM. DPPH Radical Scavenging Activity (RSA) results were expressed as the % of radical inhibition. For the ABTS assay, a 7 mM ABTS (Sigma-Aldrich, St. Louis, MO, USA) solution and a 2.45 mM potassium persulfate solution were prepared. Equal volumes of the two solutions were mixed and incubated in the dark at room temperature for 24 h to generate the ABTS•^+^ radical cation. Prior to use, the ABTS•^+^ solution was diluted with distilled water in a 2:25 (*v/v*) ratio to obtain an absorbance of 0.7 ± 0.02 at 734 nm. In a 96-well microplate, 200 µL of the diluted ABTS•^+^ solution and 20 µL of each extract were mixed and incubated at room temperature in the dark for 30 min. Absorbance was measured at 734 nm. Trolox was used as the standard, with a calibration curve prepared using concentrations from 0 to 850 µM. Results were expressed as µmol Trolox equivalents (TE) per gram of DW. Both protocols were adapted for microplate use from Jafri SAA et al. [67].

### 4.8. Determination of Phenolic Composition

To determine the phenolic composition of the extracts, a HPLC analysis was performed using an Agilent 1200 HPLC system (Agilent, Palo Alto, CA, USA) equipped with an Agilent ZORBAX Eclipse XDB-C18 column (4.6 mm D × 150 mm L, particle size 5 µm) and a diode array detector (DAD). HPLC-DAD conditions followed the protocol described by [68] with some modifications. The column temperature was maintained at 30 °C, with a sample injection volume of 10 µL, and a constant flow rate of 1 mL/min. The mobile phase consisted of water with 1% acetic acid (A) and acetonitrile with 1% acetic acid (B), and the gradient conditions were 0–5 min 95–70% A; 5–10 min 70–65% A; 10–20 min 65–50% A; 20–25 min 50–10% A. The detector was adjusted at 250, 280, and 330 wavelengths (nm), and the identification of phenolic compounds was performed by matching the retention time of chromatogram peaks with those of available phenolic standards. These were gallic acid, catechin, caffeic acid, epicatechin, rutin, ferulic acid, and naringenin (Sigma-Aldrich, St. Louis, MO, USA). Calibration curves for each standard were used to determine the samples’ phenolic content (0.30–12.50 µg). Individual peaks that did not match any of the standards were quantified as equivalents of the closest standard.

### 4.9. Statistical Analysis

A total of 20 explants were cultured in each treatment, with five replicates per medium, and 4 explants per culture vessel. The shooting and rooting experiments each included seven treatments, with 140 explants per experiment. A total of 60 plants were used to evaluate hardening and acclimatization. For phytochemical characterization, all experiments were performed in triplicate.

The data were statistically analyzed using IBM SPSS Statistics 30.0 software (SPSS Inc., Chicago, IL, USA). All the results are expressed as the mean ± standard error (SE). A one-way analysis of variance (ANOVA) was performed to test differences among in vitro treatments and samples from each ex vitro period. The means were compared for significance of differences using Duncan’s test at *p*  ≤  0.05. Pearson’s correlation analysis was performed to evaluate the correlation between the total phenolic content, total flavonoid content, and the antioxidant activities of extracts from *A. ludoviciana* samples.

## 5. Conclusions

In this study, a micropropagation protocol for *Artemisia ludoviciana* using shoot segments as initial explants was successfully established. In vitro shoot and root proliferation followed by hardening and acclimatization under ex vitro conditions was achieved, resulting in well-developed, morphologically uniform plants. This protocol supports the conservation, sustainable use, and large-scale propagation of the species. Furthermore, the evaluation of the phenolic content and antioxidant activity of micropropagated plants at different ex vitro stages revealed variations in secondary metabolite levels. These variations showed that the accumulation of phenolic and flavonoid compounds in *A. ludoviciana* is regulated by developmental and physiological factors. Comparisons with the mother plant showed that micropropagated plants maintained their morphological and phytochemical traits, supporting their use for medicinal and research purposes. These findings could help determine the optimal time to harvest the plant, as some compounds were more concentrated at certain ex vitro stages. The development of micropropagated plants will continue to be monitored, providing an opportunity for future research. Finally, DNA barcoding of *A. ludoviciana* is also needed to assess the genetic stability of in vitro-propagated plants and the genetic diversity of the species.

## Figures and Tables

**Figure 1 plants-14-03781-f001:**
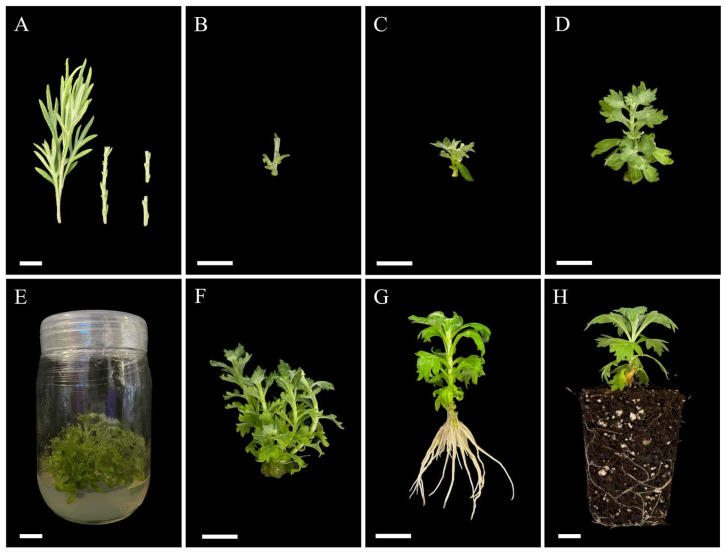
Micropropagation of *A. ludoviciana* plantlets (**A**–**E**). Initial explant sections utilized for the induction phase (**A**). Shoot induction after 7 days (**B**), 14 days (**C**), and 28 days (**D**) of culture on Murashige and Skoog (MS) media containing 6-benzylaminopurine (BAP) (0.05 mg L^−1^); shoot multiplication after 4 weeks of culture on MS media containing BAP (0.1 mg L^−1^) (**E**,**F**); rooting after 2 weeks of culture on ½ MS media containing indoleacetic acid (IAA) (0.5 mg L^−1^) (**G**); and hardening (**H**). Scale bar: 1 cm.

**Figure 2 plants-14-03781-f002:**
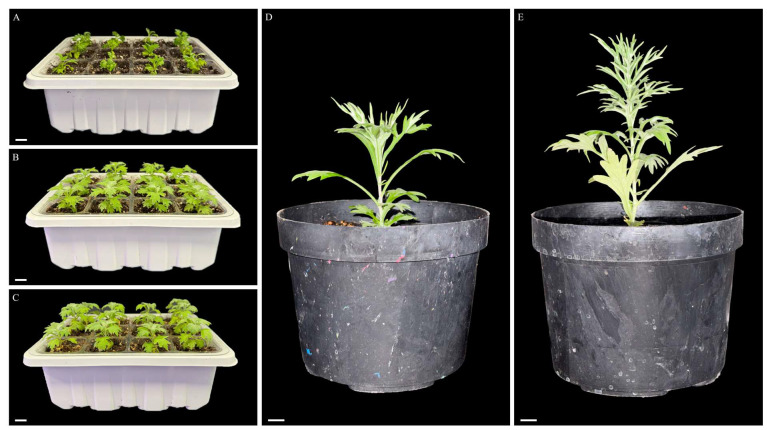
Hardening and acclimatization of *A. ludoviciana* plantlets (**A**–**E**). Hardening at 0 days (**A**), 7 days (**B**), and 14 days (**C**) after in vitro propagation; Plant after 1 month (**D**) and 2 months (**E**) after acclimatization. Scale bar: 1 cm.

**Figure 3 plants-14-03781-f003:**
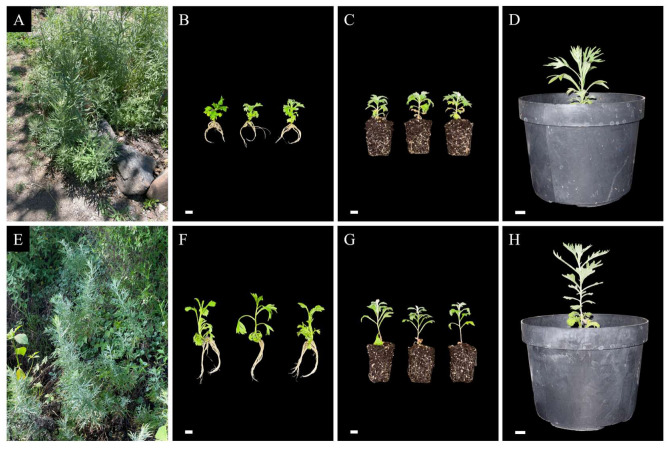
Micropropagation of two different *A. ludoviciana* populations (**A**–**H**). Population from the Botanical Garden of the Autonomous University of Queretaro, Queretaro (**A**–**D**). Population from La Cañada, Queretaro (**E**–**H**). Wild mother plants (**A**,**E**); Plantlets after in vitro propagation (**B**,**F**); Plantlets after 14 days of hardening (**C**,**G**). 1-month-acclimatized plants (**D**,**H**). Scale bars: 1 cm.

**Table 1 plants-14-03781-t001:** Effect of Different Cytokinins Concentrations on shoot multiplication from In vitro Nodal Segments of *Artemisia ludoviciana.*

Treatment	BAP ^1^ (mg/L)	KIN ^2^ (mg/L)	No. Shoots/ Explant *	Shoot Length (cm) *
T0	0.00	0.00	1.20 ± 0.09 ^c^	1.01 ± 0.08 ^d^
T1	0.10	0.00	3.30 ± 0.34 ^a^	3.00 ± 0.12 ^a^
T2	0.50	0.00	2.45± 0.20 ^b^	0.98 ± 0.07 ^d^
T3	1.00	0.00	2.60 ± 0.18 ^b^	0.68 ± 0.03 ^e^
T4	0.00	0.10	1.55 ± 0.14 ^c^	0.95 ± 0.08 ^d^
T5	0.00	0.50	2.20 ± 0.24 ^b^	1.75 ± 0.09 ^b^
T6	0.00	1.00	2.35 ± 0.17 ^b^	1.34 ± 0.08 ^c^

* The results are reported as mean ± standard error (SE). Values with different letters in a column are significantly different, according to Duncan’s test (*p* < 0.05). The data were recorded after a four-week culture. ^1^ BAP 6-benzylaminopurine; ^2^ KIN Kinetin.

**Table 2 plants-14-03781-t002:** Effect of different auxin concentrations on root induction from in vitro shoots of *Artemisia ludoviciana.*

Treatment	NAA ^1^ (mg/L)	IAA ^2^ (mg/L)	No. Roots/Shoot *	Root Length (cm) *	Shoot Length (cm) *
R0	0.00	0.00	8.70 ± 0.26 ^c^	3.44 ± 0.09 ^d^	2.35 ± 0.08 ^b^
R1	0.10	0.00	9.60 ± 0.43 ^bc^	3.71 ± 0.08 ^bc^	2.44 ± 0.08 ^a^
R2	0.50	0.00	10.60 ± 0.69 ^b^	4.04 ± 0.06 ^a^	2.20 ± 0.04 ^bc^
R3	1.00	0.00	14.30 ± 0.92 ^a^	3.78 ± 0.09 ^b^	2.28 ± 0.06 ^abc^
R4	0.00	0.10	10.55 ± 0.26 ^b^	3.48 ± 0.10 ^cd^	2.43 ± 0.07 ^a^
R5	0.00	0.50	14.45 ± 0.56 ^a^	2.85 ± 0.05 ^e^	2.31 ± 0.06 ^abc^
R6	0.00	1.00	13.85 ± 0.44 ^a^	2.66 ± 0.11 ^e^	2.11 ± 0.06 ^d^

* The results are reported as mean ± standard error (SE). Values with different letters in a column are significantly different, according to Duncan’s test (*p* < 0.05). The data were recorded after a two-week culture. ^1^ NAA α-naphthaleneacetic acid; ^2^ IAA Indoleacetic acid.

**Table 3 plants-14-03781-t003:** Dry weight percentage, total phenolic content (TPC), total flavonoid content (TFC), radical scavenging activity (RSA), and Trolox equivalent antioxidant capacity (TEAC) from *A. ludoviciana* aerial parts at different ex vitro stages and the control plant.

Sample	Dry Weight (%)	TPC (g GAE/100 g DW)	TFC (g QE/100 g DW)	RSA (Inhibition %)	TEAC (µmol TE/g DW)
IP	8.00 ± 0.11 ^f^	1.44 ± 0.00 ^f^	0.29 ± 0.09 ^d^	59.83 ± 3.70 ^d^	71.28 ± 2.68 ^d^
D7	10.52 ± 0.25 ^e^	1.34 ± 0.01 ^e^	0.13 ± 0.06 ^d^	38.21 ± 1.08 ^c^	58.51 ± 2.29 ^e^
D14	17.97 ± 0.17 ^d^	2.50 ± 0.01 ^c^	0.91 ± 0.03 ^c^	81.40 ± 0.19 ^a^	85.56 ± 1.50 ^c^
1M	21.28 ± 0.56 ^c^	1.80 ± 0.00 ^d^	0.96 ± 0.05 ^c^	82.50 ± 0.79 ^a^	108.61 ± 5.19 ^b^
2M	26.39 ± 0.77 ^b^	2.63 ± 0.01 ^b^	1.38 ± 0.12 ^b^	81.92 ± 0.50 ^a^	118.51 ± 1.35 ^b^
MP	31.84 ± 0.70 ^a^	3.10 ± 0.03 ^a^	1.84 ± 0.08 ^a^	84.21 ± 0.38 ^a^	152.04 ± 5.78 ^a^

The results are reported as mean ± standard error (SE) of three replicates. Values with different letters in the same column are significantly different, according to Duncan’s test (*p* < 0.05). GAE: Gallic acid equivalent, QE: Quercetin equivalent, TE: Trolox equivalent, DW: Sample dry weight, IP: In vitro plants, D7: Plants after 7 days of hardening, D14: Plants after 14 days of hardening, 1M: Plants after 4 weeks of acclimatization, 2M: Plants after 8 weeks of acclimatization, and MP: Mother plant.

**Table 4 plants-14-03781-t004:** Pearson’s correlation coefficients between the total phenolic content (TPC), total flavonoid content (TFC), radical scavenging activity (RSA), and Trolox equivalent antioxidant capacity (TEAC) of extracts obtained from *A. ludoviciana* aerial parts at different ex vitro stages and the control plant.

	TPC	TFC	RSA	TEAC
**T** **PC**	1			
**TFC**	0.942 **	1		
**RSA**	0.783	0.841 *	1	
**TEAC**	0.853 *	0.974 **	0.790	1

** Significant correlation at <0.01; * Significant correlation at <0.05.

**Table 5 plants-14-03781-t005:** HPLC-DAD phenolic identification and quantification of extracts obtained from *A. ludoviciana* aerial parts at different ex vitro stages and the control plant.

Compound (mg/g DW)	IP	D7	D14	1M	2M	MP
Peak 1	0.65 ± 0.02 ^a^	0.62 ± 0.01 ^b^	0.56 ± 0.00 ^d^	0.59 ± 0.00 ^c^	0.56 ± 0.00 ^d^	0.58 ± 0.01 ^cd^
Peak 2	0.42 ± 0.03 ^d^	0.49 ± 0.03 ^d^	3.93 ± 0.59 ^b^	2.48 ± 0.08 ^c^	4.59 ± 0.25 ^b^	8.04 ± 0.73 ^a^
Caffeic acid	-	-	0.20 ± 0.02 ^d^	0.33 ± 0.01 ^b^	0.56 ± 0.01 ^a^	0.28 ± 0.02 ^c^
Peak 4	-	-	-	-	-	0.13 ± 0.01
Peak 5	0.24 ± 0.03 ^a^	0.32 ± 0.06 ^a^	-	-	-	0.05 ± 0.03 ^b^
Peak 6	-	-	0.18 ± 0.03 ^c^	0.52 ± 0.02 ^b^	0.65 ± 0.03 ^a^	0.72 ± 0.05 ^a^
Ferulic acid	1.94 ± 0.19 ^c^	2.15 ± 0.12 ^c^	4.46 ± 0.30 ^b^	4.44 ± 0.08 ^b^	4.63 ± 0.11 ^ab^	5.04 ± 0.17 ^a^
Peak 8	-	-	0.13 ± 0.02 ^d^	0.51 ± 0.01 ^c^	1.43 ± 0.11 ^b^	2.27 ± 0.25 ^a^
Peak 9	6.21 ± 0.16 ^c^	7.65 ± 0.33 ^b^	4.80 ± 0.18 ^d^	10.34 ± 0.09 ^a^	4.54 ± 0.35 ^d^	5.75 ± 0.26 ^c^
Peak 10	-	0.32 ± 0.06 ^ab^	-	0.44 ± 0.01 ^a^	0.05 ± 0.04 ^c^	0.29 ± 0.07 ^b^

The results are reported as mean ± standard error (SE) of three replicates. Values with different letters in the same file are significantly different, according to Duncan’s test (*p* < 0.05). DW: Sample dry weight, IP: In vitro plants, D7: Plants after 7 days of hardening, D14: Plants after 14 days of hardening, 1M: Plants after 4 weeks of acclimatization, 2M: Plants after 8 weeks of acclimatization, and MP: Mother plant.

**Table 6 plants-14-03781-t006:** Shoot multiplication treatments tested on nodal explants of *Artemisia ludoviciana.*

Treatment	BAP ^1^ (mg/L)	KIN ^2^ (mg/L)
T0	0.00	0.00
T1	0.10	0.00
T2	0.50	0.00
T3	1.00	0.00
T4	0.00	0.10
T5	0.00	0.50
T6	0.00	1.00

^1^ BAP (6-Benzylaminopurine); ^2^ KIN (Kinetin).

**Table 7 plants-14-03781-t007:** Rooting treatments tested on in vitro shoots of Artemisia ludoviciana.

Treatment	NAA ^1^ (mg/L)	IAA ^2^ (mg/L)
R0	0.00	0.00
R1	0.10	0.00
R2	0.50	0.00
R3	1.00	0.00
R4	0.00	0.10
R5	0.00	0.50
R6	0.00	1.00

^1^ NAA (1-Naphthaleneacetic acid); ^2^ IAA (Indole-3-acetic acid).

## Data Availability

The original contributions presented in this study are included in the article. Further inquiries can be directed to the corresponding author.

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
