# Peer review of "Micropropagation and Phytochemical Characterization of Artemisia ludoviciana Nutt.: Antioxidant Activity and Phenolic Profiles"

_plants, 2025, doi:10.3390/plants14243781_

Round 1

Reviewer 1 Report

Comments and Suggestions for Authors

The article Micropropagation and Phytochemical Characterization of Artemisia ludoviciana Nutt.: Phenolic Profiles and Antioxidant Activity is relevant and interesting to readers. The phytochemical characterization is the differential of this study. The phytochemical characterization is the distinguishing feature of this study. There are some corrections in Materials and Methods. Please expand the discussion on genetic variability with somaclonal variation. 

Author Response

1. Summary

We thank the reviewer for his thoughtful and constructive comments on our manuscript. We have made corrections in the materials and methods and carefully considered your suggestions and expanded the discussion on genetic variability and somaclonal variation in the revised manuscript. Detailed responses to all specific comments, along with the corresponding corrections, are provided below and highlighted in the track-changed version of the manuscript.

2. Answers to the reviewer

Comments 1: Is it epigenetic variations?

Response 1: These changes are mostly related to adaptive and developmental processes, which are necessary for survival outside of the in vitro conditions. We have highlighted this information in lines 379–387.

Comments 2: Could you relate these results to somaclonal and epigenetic variation? Genetic stability is an important Result. I suggest increase this point on Discussion.

Response 2: These results are related to adaptive and developmental changes in the plant, as well as genetic differences between both populations. It is important to consider during micropropagation. We increased this point on the Discussion as suggested (lines 369-378), to highlight the importance of genetic stability.

Comments 3: Hormone or PGR? Please review the entire text.

Response 3. We have changed hormone to PGR in the entire document

Comments 4: R5

Response 4: correction has been applied

Comments 5: Did you compare genetic variability? Please include.

Response 5: We did not compare genetic variability in the present study, since this was not the primary focus of the present research article which  is the in vitro micropropagation and phytochemical characterization.

Reviewer 2 Report

Comments and Suggestions for Authors

This paper discusses the micropropagation, phenolic composition, and antioxidant activity of A. ludoviciana. The authors tested seven different micropropagation media and seven different rooting media. They selected the best media for micropropagation and rooting, then acclimatised 20 plants. From these 20 plants, samples were collected at five time points after in vitro micropropagation: immediately post-propagation (IP), after 7 days (D7) and 14 days (D14) of hardening, and after 4 weeks (1M) and 8 weeks (2M) of acclimatisation. Biomass, dry weight, total phenolic content (TPC), total flavonoid content (TFC), and antioxidant activity (DPPH and ABTS assays) were measured. It is a pity that the authors did not identify the most important chemical in Artemisia species—artemisinin. This would have added relevance and interest to the paper. However, this work may be relevant to those working with Artemisia species and to scientists interested in changes in phytochemical content during in vitro and ex vitro experiments. 

Please find Specific comments below:

Line 16 and Line 67:  ´this research provides a fast and reliable micropropagation protocol for this plant through direct organogenesis´ and ´micropropagation protocol for A. ludoviciana through direct shoot and root organogenesis´

Comment: In this article, a micropropagation protocol is described, not an organogenesis protocol.

Line 21/22: Moreover, 1/2 MS media supplemented with 0.5 mg L⁻¹ indole-3-acetic acid improved the number (14.45 ± 0.56) and quality of roots.

Comment: This data is not supported in Table 2.

Line 156: The highest root induction was obtained at 0.5 mg L⁻¹ IAA (R5), with 14.45 ± 0.56 roots per shoot.

Comment: This contradicts Table 2, which shows that treatment R5 produced 2.20 ± 0.24 roots per shoot.

Line 167: The meaning of the column "No. shoots/roots" in Table 2 is unclear. Does it refer to the number of shoots per root, or the number of roots per shoot?

Line 433:  In this type of measurement superscript should be used µmol m−2 s−1. Comment: Make this change in this sentence and throughout the paper.

Line 449: Please clarify which is the pre-induction medium: MS hormone-free or MS + 0.05 mg/L BAP?

Line 463: Please explain why you used 1.5 cm long shoots when you had 3 ± 0.12 cm (Table 1) long shoots from the T1 treatment.

Line 472: 0.5 mg L⁻¹ IAA treatment should be abbreviated R5 and not T5

Line 487: The abbreviation under the table is incorrect (KIN instead of IAA). Table 6 should be placed under the title "Rooting".

Line 501: ´Total phenolic and flavonoid content and antioxidant capacity´

Comment: There is no data on the amount of plant tissue (in grams of fresh and dried plant tissue) you used for these experiments.

Line 503: ´Samples were taken after in vitro micropropagation´

Comment: Please specify exactly when the samples were taken. Was it after in vitro propagation, on the last day of in vitro propagation, or at another time?

Line 505: How many acclimatised plants did you use for plant tissue sampling in each term (IP, D7, D14, M1, M2) to calculate total biomass? How many samples did you weigh? Please provide data on fresh biomass and dry matter mass of samples in the Supplementary table.

Line 519: ¨ 200 L of the reaction were added in a 96-well microplate¨

Comment: Is this the correct unit?

Line 573: ´total of 20 explants were cultured in each treatment, with five replicates per medium, and 4 explants per culture vessel (n = 140).

Comments: Since you had seven treatments for shooting and seven treatments for rooting, there should be 140 explants for shooting and 140 explants for rooting. Please specify how many explants were used in the shooting and rooting experiments.

Please specify the number of acclimatised plants. If I have counted correctly, it should be 20.

Please specify the number of samples used for biomass calculations.

Please specify the number and mass of samples used for TPC, TFC, DPPH, and ABTS assays.

Line 576: A one-way analysis of variance (ANOVA) was performed, and the means were compared for significance of differences using Duncan’s  test at p ≤ 0.05.

Comment: You should specify what you tested with ANOVA.

Author Response

1. Summary

We sincerely thank the reviewer for their careful evaluation and constructive feedback on our manuscript. We have thoroughly revised the text to address all comments, ensuring consistency between the results and tables, clarifying experimental details, and correcting terminology, abbreviations, and units throughout. Additional explanations have been included regarding sampling procedures, the number of explants, and the statistical analyses performed. All specific revisions and corresponding corrections are detailed below and highlighted in the track-changed version of the revised manuscript.

2. Answers to the reviewer

Comments 1: Line 16 and Line 67:  ´this research provides a fast and reliable micropropagation protocol for this plant through direct organogenesis´ and ´micropropagation protocol for A. ludoviciana through direct shoot and root organogenesis´ In this article, a micropropagation protocol is described, not an organogenesis protocol.

Response 1: We would like to clarify that the micropropagation protocol developed in this study was achieved via direct organogenesis, as shoots and roots formed directly from the explant tissue without any undifferentiation phase. This characteristic meets the definition of direct organogenesis commonly used in plant tissue culture.

Comments 2: Line 21/22: Moreover, 1/2 MS media supplemented with 0.5 mg L⁻¹ indole-3-acetic acid improved the number (14.45 ± 0.56) and quality of roots. This data is not supported in Table 2.

Response 2: We have corrected Table 2 so that the column now accurately reflects the results described in the text. Specifically, the number and quality of roots (14.45 ± 0.56) for 1/2 MS media supplemented with 0.5 mg L⁻¹ indole-3-acetic acid are now properly presented. We apologize for the oversight and thank the reviewer for bringing it to our attention.

Comments 3: Line 156: The highest root induction was obtained at 0.5 mg L⁻¹ IAA (R5), with 14.45 ± 0.56 roots per shoot. This contradicts Table 2, which shows that treatment R5 produced 2.20 ± 0.24 roots per shoot.

Response 3: This comment is related to the previous observation regarding Table 2. We have corrected the table so that the values now accurately reflect the results described in the text. We apologize for the earlier inconsistency.

Comments 4: Line 167: The meaning of the column "No. shoots/roots" in Table 2 is unclear. Does it refer to the number of shoots per root, or the number of roots per shoot?

Response 4: The column in Table 2 refers to the number of roots per shoot, so we have corrected the column header to “No. roots/shoot” to accurately reflect the data. This change ensures consistency with the text and clarifies the intended meaning

Comments 5: NLine 433:  In this type of measurement superscript should be used µmol m−2 s−1. Comment: Make this change in this sentence and throughout the paper.

Response 5: We have corrected all superscripts throughout the manuscript so that they are now properly formatted.

Comments 6: Line 449: Please clarify which is the pre-induction medium: MS hormone-free or MS + 0.05 mg/L BAP?.

Response 6: We appreciate this observation and have clarified in the text that the pre-induction medium is full-strength MS supplemented with 0.05 mg L⁻¹ BAP. This ensures that the methodology is now clearly described and reproducible for the reader.

Comments 7: Line 463: Please explain why you used 1.5 cm long shoots when you had 3 ± 0.12 cm (Table 1) long shoots from the T1 treatment.

Response 7: Although the shoots could be rooted at various lengths, we chose 1.5 cm long shoots to maintain better control over the starting size and growth of the explants. This length provided a consistent starting point and sufficient margin to ensure uniformity when transferred to the rooting media.

Comments 8: Line 472: 0.5 mg L⁻¹ IAA treatment should be abbreviated R5 and not T5

Response 8: we have corrected the treatment abbreviation so that 0.5 mg L⁻¹ IAA is now properly labeled as R5 in the manuscript.

Comments 9: Line 487: The abbreviation under the table is incorrect (KIN instead of IAA). Table 6 should be placed under the title "Rooting".

Response 9: Table 6 has been corrected so that the abbreviation now reads IAA, and the table is properly placed under the title “Rooting.” The corrected table is now titled: “Rooting treatments tested on in vitro shoots of Artemisia ludoviciana.”

Comments 10: Line 501: ´Total phenolic and flavonoid content and antioxidant capacity´ There is no data on the amount of plant tissue (in grams of fresh and dried plant tissue) you used for these experiments.

Response 10: To address this point, we have revised the Methods section to specify the exact amount of plant material used in the extraction procedure. In our experiments, 1 g of dried plant tissue was extracted with 10 mL of solvent (1:10 w/v). This clarification has been incorporated to enhance the methodological transparency and reproducibility of the study (Lines 510-513).

Comments 11: Line 503: ´Samples were taken after in vitro micropropagation´

Comment: Please specify exactly when the samples were taken. Was it after in vitro propagation, on the last day of in vitro propagation, or at another time?

Response 11: We have corrected the text to clarify when the samples were taken. Aerial parts were collected on the last day of in vitro micropropagation (IP), after 7 days (D7) and 14 days (D14) of hardening, and at 4 weeks (1M) and 8 weeks (2M) of acclimatization.

Comments 12: Line 505: How many acclimatised plants did you use for plant tissue sampling in each term (IP, D7, D14, M1, M2) to calculate total biomass? How many samples did you weigh? Please provide data on fresh biomass and dry matter mass of samples in the Supplementary table.

Response 12:. We would like to clarify that the term “biomass” was not properly employed in the original manuscript. Instead, we measured the dry weight percentage to monitor changes in the plant material as it transitioned from in vitro to ex vitro conditions. During acclimatization, exposure to lower relative humidity caused the plants to shift from a high-water, low-dry-mass state to a lower-water, higher-dry-mass state. This measurement reflects the adjustment of the plants’ water content and dry matter accumulation over time. Corrections regarding this terminology have been made in the manuscript to improve clarity.

Comments 13: Line 519: ¨ 200 L of the reaction were added in a 96-well microplate¨

Comment: Is this the correct unit?

Response 13:  We have revised the text to indicate the correct volume, which is 200 µL. This correction ensures accuracy and clarity in the methodological description.

Comments 14: Line 573: ´total of 20 explants were cultured in each treatment, with five replicates per medium, and 4 explants per culture vessel (n = 140).’ Since you had seven treatments for shooting and seven treatments for rooting, there should be 140 explants for shooting and 140 explants for rooting. Please specify how many explants were used in the shooting and rooting experiments. Please specify the number of acclimatised plants. If I have counted correctly, it should be 20. Please specify the number of samples used for biomass calculations. Please specify the number and mass of samples used for TPC, TFC, DPPH, and ABTS assays.

Response 14: We have improved the text to clarify the experimental design and statistical analysis (Lines 580-584). Furthermore, relevant details have been added to the methodology sections to enhance transparency, reproducibility, and accuracy in the description of the experiments. This clarification is important because it allows readers to better understand the experimental setup and methodology, ensuring the results can be properly interpreted and the study can be reproduced.

Comments 15: Line 576: A one-way analysis of variance (ANOVA) was performed, and the means were compared for significance of differences using Duncan’s  test at p ≤ 0.05. You should specify what you tested with ANOVA.

Response 15: We have clarified in the manuscript that a one-way ANOVA was performed to test differences among in vitro treatments and samples from each ex vitro period, which is important to ensure the statistical comparisons are clear and the results can be properly interpreted.

3. Additional comments.

We sincerely thank the reviewer for their very valuable comments, which have greatly contributed to improving the clarity, accuracy, and overall quality of the manuscript.

Reviewer 3 Report

Comments and Suggestions for Authors

The study is well designed, the article is well written and also well documented with high quality photographs. I have only 2 suggestions; for the readers who are not familiar with the plant propagation protocols it might be helpful to specify whether surface sterilization was performed by submerging the plants in the antimicrobial solutions or applying solutions using some other technique. The other suggestion is to explain in the discussion how is decided to use the specific protocol and whether there were some trial and errors that might be helpful for the readers.

Author Response

1. Summary

We sincerely thank the reviewer for their thoughtful suggestions on our manuscript. We appreciate the positive evaluation of our study design, writing, and documentation. In response to your comments, we gladly followed your advice and clarified the surface sterilization procedure in the Materials and Methods section, and expanded the Discussion section to explain why we chose this specific micropropagation protocol.

2. Answers to the reviewer

Comments 1: For the readers who are not familiar with the plant propagation protocols it might be helpful to specify whether surface sterilization was performed by submerging the plants in the antimicrobial solutions or applying solutions using some other technique.

Response 1: We have clarified the surface sterilization procedure in the Materials and Methods (Line 350). Specifically, the explants were submerged in the antimicrobial solutions for the indicated time periods, ensuring effective sterilization before culture.

Comments 2: The other suggestion is to explain in the discussion how is decided to use the specific protocol and whether there were some trial and errors that might be helpful for the readers.

Response 2: We have expanded the Discussion to explain why the specific micropropagation protocol was used (Lines 242-244).

Reviewer 4 Report

Comments and Suggestions for Authors

Title

Micropropagation and Phytochemical Characterization of Artemisia ludoviciana Nutt.: Phenolic Profiles and Antioxidant Activity

Review the order of attributions

Abstract

Line 15: Avoid subjective words like "fast" and "reliable."

Line 19: What medium?

Line 19: Is the concentration 0.05 mg L⁻¹ correct?

Lines 19-20: What is the purpose of this sentence? Establishment.

Lines 25-27: Conducting studies for the first time is not justification for publication.

Overall, the abstract lacks a good justification. Why this plant? Why use this type of in vitro propagation? Does seed or asexual propagation not work?

Keywords

Do not repeat words in the title.

Introduction

Line 36: Citations are linked [1,2]; review the entire text.

Line 39: What does this line contribute to the study?

Line 63: What are these limitations?

The introduction is short and based on the medicinal properties of this plant. It is necessary to mention in detail the limitations of conventional propagation; this justifies the use of in vitro techniques.

Results

In lines 77-78, it is not necessary to mention the disinfection treatment if only one treatment was used.

In lines 79-81, this is called bud proliferation; could you include a section on it?

In lines 124: "Bud breaking"? Shoot formation?

In lines 124-131, I think it would be better to describe the results by column of the table (each variable). First, give a general overview, then the results.

2.3. Root induction should be arranged in the same way.

Table 2: Why are the treatments named R1, R2, etc.? The treatments above should be M1, M2, etc.

Lines 204-207: Generally, experiments are performed in duplicate and triplicate (experimental design); it is not necessary to state this in the text.

Table 3: I wonder if the variables evaluated are directly proportional to biomass content? That is, as the plant grows, there is more biomass and, consequently, more compounds.

The results are particularly interesting; however, they belong to a routine technical report when performing a plant micropropagation protocol, despite the lack of information in the literature.

Materials and methods

Line 432 Why was this photoperiod used?

Line 433 superscripts.

Line 442 What size were the nodal segments? How many axillary buds did they contain?

Line 450 Did you have axillary buds?

Line 432 Why was this treatment used only at this stage?

The materials and methods should provide as much detail as possible to be reproducible.

Discussion

Lines 327-333: Information is too general.

Lines 337-338: I think it's better to first mention what was observed in the presented study and then discuss similar studies.

In general, the discussion needs to be improved; only previous studies where experimental conditions are equal can be discussed. The discussion focuses on whether the use of cytokinins generates outbreaks in the study species; this is something expected even before conducting the study. Just as the use of auxins generates roots, these topics should be explored in more depth. In the presented study, it was observed that low concentrations of BAP generated the greatest number of shoots, and a hormetic effect could be observed, etc. Hormesis indicates that...

Conclusion

Should be consistent with the objective of the study, and should be improved.

Author Response

1. Summary

We thank the reviewer for their thoughtful and constructive comments, which greatly helped us improve the quality of the manuscript. We have carefully revised the text to address all suggestions, ensuring greater clarity, consistency, and precision throughout. Additional methodological details were included where needed to enhance reproducibility. The discussion was expanded to provide a more comprehensive interpretation of the results in the context of previous studies and what is reported in the literature, thereby strengthening the overall scientific relevance of the manuscript.

2. Answers to the reviewer

Comments 1: Review the order of attributions. (Title)

Response 1: The order of attributions in the title has been revised accordingly. The new title now reads: “Micropropagation and Phytochemical Characterization of Artemisia ludoviciana Nutt.: Antioxidant Activity and Phenolic Profiles” to be consistent with the content and presentation in the manuscript.

Comments 2: Line 15: Avoid subjective words like "fast" and "reliable."

Response 2: We have made the necessary corrections throughout the text to avoid the use of subjective words such as “fast” and “reliable.”

Comments 3: Line 19-20: What medium? Is the concentration 0.05 mg L⁻¹ correct? What is the purpose of this sentence? Establishment.

Response 3:. The sentence regarding establishment, including the medium and concentration, was removed from the abstract to focus on more relevant aspects of the study.

Comments 4: Lines 25-27: Conducting studies for the first time is not justification for publication. Overall, the abstract lacks a good justification. Why this plant? Why use this type of in vitro propagation? Does seed or asexual propagation not work?

Response 4:. We agree that simply stating a study is conducted for the first time does not justify publication. In response, we revised the abstract to clarify the motivation behind the study, explaining the medicinal importance of Artemisia ludoviciana, the threats to its natural populations, and the limitations of conventional propagation methods compared to micropropagation.

Comments 5: Do not repeat words in the title (Keywords)

Response 5: We have updated the keywords to avoid repeating words from the title.

Comments 6: Line 36: Citations are linked [1,2]; review the entire text.

Response 6: We have corrected citations in accordance with the journal’s guidelines.

Comments 7: Line 39: What does this line contribute to the study?

Response 7:

We consider this information important because it highlights the cultural and historical significance of Artemisia ludoviciana. This emphasizes the plant’s traditional use and supports the relevance of studying its propagation and bioactive properties.

Comments 8: Line 63: What are these limitations?The introduction is short and based on the medicinal properties of this plant. It is necessary to mention in detail the limitations of conventional propagation; this justifies the use of in vitro techniques.

Response 8: We have revised the introduction to be more specific about the limitations of conventional propagation methods. These include low multiplication rates, environmental constraints, and susceptibility to pathogens. By stating these challenges, the introduction now better justifies the use of micropropagation as a faster, large-scale, and disease-free alternative, highlighting its relevance for both conservation and research purposes.

Comments 9: In lines 77-78, it is not necessary to mention the disinfection treatment if only one treatment was used.

Response 9: Although only one disinfection treatment was used, we consider it important to include this information, as it provides a detailed description of the results and offers context for each subsequent step of the micropropagation protocol.

Comments 10: In lines 79-81, this is called bud proliferation; could you include a section on it?

Response 10: While we evaluated different media for this stage, the variations observed were minimal and did not significantly affect the overall outcomes of the micropropagation protocol. We chose to focus the manuscript on the most relevant results to keep it clear and straightforward.

Comments 11: In lines 124: "Bud breaking"? Shoot formation?

Response 11: We agree that “shoot formation” is a more appropriate terminology. We made this correction to improve the accuracy and clarity of the manuscript.

Comments 12: In lines 124-131, I think it would be better to describe the results by column of the table (each variable). First, give a general overview, then the results. 2.3. Root induction should be arranged in the same way. 

Response 12:. We understand the suggestion to describe the results column by column. In the current Results section, we focused on highlighting the most relevant data, as the table itself provides a comprehensive overview of all results. In response, the discussion has been revised and strengthened to offer a thorough interpretation of the results.

Comments 13: Table 2: Why are the treatments named R1, R2, etc.? The treatments above should be M1, M2, etc.

Response 13: This nomenclature was intentionally used to indicate that these treatments correspond to the rooting phase, differentiating them from the shoot multiplication treatments. This distinction helps differentiate the stage of the micropropagation protocol being discussed and ensures readers can easily follow the experimental design.

Comments 14: Lines 204-207: Generally, experiments are performed in duplicate and triplicate (experimental design); it is not necessary to state this in the text. 

Response 14: We would like to clarify that this statement does not refer to the number of replicates, but rather to the application of the established micropropagation protocol to a different population of A. ludoviciana. It is important to include this information to demonstrate that the protocol is reproducible and can be successfully applied to other populations.

Comments 15: Table 3: I wonder if the variables evaluated are directly proportional to biomass content. That is, as the plant grows, there is more biomass and, consequently, more compounds.

Response 15: We agree that the relationship between plant growth and compound content is important to consider. We clarify that the term “biomass” was not accurately used in the original manuscript; instead, we measured the dry weight percentage to monitor changes in plant material during the transition from in vitro to ex vitro conditions. All measurements of phenolic compounds and antioxidant activity were expressed on a dry weight basis, ensuring that the observed variations are independent of differences in plant growth.

Comments 16: Line 432 Why was this photoperiod used? 

Response 16: Although the 12-hour light/12-hour dark photoperiod was used based on the fixed settings of the growth chamber, it also approximates the natural day–night cycles of this plant in the wildness (México).

Comments 17: Line 433 superscripts.

Response 17: We have corrected all superscripts throughout the manuscript so that they are now properly formatted. 

Comments 18: Line 442 What size were the nodal segments? How many axillary buds did they contain?

Response 18: The nodal segments harvested from the mother plants were approximately 1.5 cm in length. We have added that each segment contained about two axillary buds (Line 347-348).

Comments 19: Line 450 Did you have axillary buds?

Response 19: Yes, the explants contained axillary buds. Since we used nodal segments, these buds are essential for initiating shoot formation and are a key component of the micropropagation protocol.

Comments 20: Line 432 Why was this treatment used only at this stage?

Response 20:  In the Results, we mention that low cytokinin concentrations were tested for shoot induction; however, these data were not included because the variations observed were minimal. This approach allowed us to focus on the most relevant results while maintaining clarity in the manuscript.

Comments 21: Lines 327-333: Information is too general.

Response 21: We consider that this information, although general, provides an essential introduction to the discussion of results. It sets context and helps guide the reader through the subsequent detailed analysis.

Comments 22: Lines 337-338: I think it's better to first mention what was observed in the presented study and then discuss similar studies.

Response 22: We have corrected this section following your suggestion, which makes the manuscript clearer and improves the logical flow by presenting our observations first before discussing similar studies (Line 242).

Comments 23: In general, the discussion needs to be improved; only previous studies where experimental conditions are equal can be discussed. The discussion focuses on whether the use of cytokinins generates outbreaks in the study species; this is something expected even before conducting the study. Just as the use of auxins generates roots, these topics should be explored in more depth.

Response 23: We have made the suggested revisions, as incorporating these corrections will undoubtedly enhance the scientific value and depth of the manuscript (Lines 246-255 and 265-268).

Comments 24: Should be consistent with the objective of the study, and should be improved (Conclusion).

Response 24: We have revised the Conclusion to make it more consistent with the objectives of the study. The new version emphasizes the successful establishment of a micropropagation protocol, highlights the observed variations in phenolic content and antioxidant activity at different ex vitro stages, and reinforces the relevance of these findings for conservation, sustainable use, medicinal applications, and further research.

Round 2

Reviewer 2 Report

Comments and Suggestions for Authors

Dear authors,

The phrase ¨ through direct organogenesis¨ should be removed from the manuscript.

Since you have started with shoot explants with two axillary buds, it is micropropagation. Direct organogenesis refers to the regeneration of organs from various types of plant tissues, but not from pre-existing meristems.

Author Response

1. Summary

We sincerely appreciate the constructive feedback provided in their previous comments on the article, and we thank the reviewer for this insightful comment regarding the terminology used in our manuscript.

2. Answers to the reviewer

Comments 1: The phrase ¨ through direct organogenesis¨ should be removed from the manuscript.

Since you have started with shoot explants with two axillary buds, it is micropropagation. Direct organogenesis refers to the regeneration of organs from various types of plant tissues, but not from pre-existing meristems.

Response 1: After carefully reviewing the use of this term, we fully agree with the reviewer that the micropropagation system described involves axillary bud proliferation and therefore should not be considered as direct organogenesis. Accordingly, we have corrected the terminology throughout the manuscript, replacing direct organogenesis with the appropriate description (Lines 68-71, 243, 479). These corrections improve the clarity and accuracy of the manuscript, ensuring that the methodology is described correctly and follows the proper terminology.

Reviewer 4 Report

Comments and Suggestions for Authors

The submitted manuscript is an improved version of a previous amnucritus.

The title was improved.

The abstract was substantially improved.

The introduction better covered the rationale for the study.

The materials and methods section was improved, providing details for reproducibility.

For results, more precise terms were used.

The discussion was expanded and compared studies with identical experimental conditions.

In the conclusion, only the statement that it is the first study was removed; as I mentioned earlier, this is not a valid justification for publication. The conclusion ends on line 486.

Author Response

1. Summary

We sincerely thank the reviewer for the careful evaluation of our manuscript and for the constructive comments, which have helped us to further improve the clarity and quality of our article.

2. Answers to the reviewer

Comments 1: In the conclusion, only the statement that it is the first study was removed; as I mentioned earlier, this is not a valid justification for publication. The conclusion ends on line 486.

Response 1: We have corrected and revised the conclusion by removing the statement referring to this study as the first and summarizing the key findings of the study more clearly (Line 478-481). These revisions improve the conclusion by ensuring that it focuses on the most important aspects of the study.
